# Phenotypic resistant single-cell characteristics under recurring ampicillin antibiotic exposure in *Escherichia coli*

Silvia Kollerová,[1] Lionel Jouvet,[1] Julia Smelková,[1] Sara Zunk-Parras,[2] Alexandro Rodríguez-Rojas,[2] Ulrich K. Steiner[1,2]

**ABSTRACT** Non-heritable, phenotypic drug resistance toward antibiotics challenges antibiotic therapies. Characteristics of such phenotypic resistance have implications for the evolution of heritable resistance. Diverse forms of phenotypic resistance have been described, but phenotypic resistance characteristics remain less explored than genetic resistance. Here, we add novel combinations of single-cell characteristics of phenotypic resistant *E. coli* cells and compare those to characteristics of susceptible cells of the parental population by exposure to different levels of recurrent ampicillin antibiotic. Contrasting expectations, we did not find commonly described characteristics of phenotypic resistant cells that arrest growth or near growth. We find that under ampicillin exposure, phenotypic resistant cells reduced their growth rate by about 50% compared to growth rates prior to antibiotic exposure. The growth reduction is a delayed alteration to antibiotic exposure, suggesting an induced response and not a stochastic switch or caused by a predetermined state as frequently described. Phenotypic resistant cells exhibiting constant slowed growth survived best under ampicillin exposure and, contrary to expectations, not only fast-growing cells suffered high mortality triggered by ampicillin but also growth-arrested cells. Our findings support diverse modes of phenotypic resistance, and we revealed resistant cell characteristics that have been associated with enhanced genetically fixed resistance evolution, which supports claims of an underappreciated role of phenotypic resistant cells toward genetic resistance evolution. A better understanding of phenotypic resistance will benefit combatting genetic resistance by developing and engulfing effective anti-phenotypic resistance strategies.

**IMPORTANCE** Antibiotic resistance is a major challenge for modern medicine. Aside from genetic resistance to antibiotics, phenotypic resistance that is not heritable might play a crucial role for the evolution of antibiotic resistance. Using a highly controlled microfluidic system, we characterize single cells under recurrent exposure to antibiotics. Fluctuating antibiotic exposure is likely experienced under common antibiotic therapies. These phenotypic resistant cell characteristics differ from previously described phenotypic resistance, highlighting the diversity of modes of resistance. The phenotypic characteristics of resistant cells we identify also imply that such cells might provide a stepping stone toward genetic resistance, thereby causing treatment failure.

**KEYWORDS** single-cell, microfluidics, antibiotic resistance, β-lactam antibiotics, phenotypic resistance, persistence, tolerance

Resistance to antibiotics, in combination with slow progress in discovering novel antibiotics, poses a major challenge to modern medicine (1). Antibiotic-resistant cells survive bactericidal concentrations of antibiotics through heritable genetically fixed mutations that include mechanisms such as decreased uptake of antibiotics, increased efflux of antibiotics, or inactivation of antibiotics (2). In consequence, resistant cells can

Address correspondence to Ulrich K. Steiner, ulrich.steiner@fu-berlin.de.

The authors declare no conflict of interest.

See the funding table on p. 16.

grow and replicate under antibiotic exposure, which results in an increased minimum inhibitory concentration (MIC). By contrast, phenotypic resistant cells are in a non-heritable drug-resistant phenotypic state that allows them to survive periods of bactericidal concentration of antibiotics—classically by non- or very slow growth prior and during exposure to the drug (3–6). The MIC of phenotypic resistant cell populations is not increased compared to susceptible populations, as periods of exposure are not limited when evaluating the MIC, and phenotypic resistant cells are usually growth arrested and therefore "inhibited." Phenotypic resistance is a form of resistance that can cause therapeutic failure of antibiotic treatment. It has a pivotal role in recurrent infections, caused by relapsing persistent infection due to incomplete eradication of pathogenic phenotypic resistant cells (7–9).

Various types of phenotypic resistance have been described including antibiotic perseverance, antibiotic tolerance, antibiotic persistence, dormancy, and states of viable but not culturable cells (5, 8–11). As these types are not always mutually exclusive, we use phenotypic resistance in a broad sense of non-heritable states that allow cells to survive or grow over periods of normally bactericidal concentrations of antibiotics. Distinction among types of phenotypic resistance hinges on population or subpopulation distinction, and growth or survival prior-, during, or post-antibiotic exposure under different concentrations of antibiotics. For instance, identifying persister cells is usually done through a characteristic biphasic killing curve observed at the population level. Only the small subpopulation of persister cells survives the above MIC concentration of antibiotics, while the majority of the susceptible population dies quickly (3, 12). By contrast, populations of tolerant cells do not show a biphasic killing curve as they are assumed to be a population-wide response and not a subpopulation one. Still, heterotolerance exists when susceptible and tolerant cells co-occur in a population, and persister cells have been described as a special case of heterotolerance. Tolerant and persister cell populations show a prolonged minimum duration of killing (MDK), but share the same MIC as susceptible populations (3–6, 11, 12). Since our work focuses on single cells, from a clonal population, under a homogeneous environment, and one antimicrobial exposure, regardless of the molecular mechanism, we adhere here to the more inclusive term of phenotypic resistance and detail observed *E. coli* growth and survival at the single-cell level.

Most phenotypic resistant cells seem to be non- or very slowly growing prior to and during antibiotic exposure, but exceptions exist (10, 12–18). Post-antibiotic exposure phenotypic resistant cells usually regain normal growth after a few hours, but again exceptions exist for cells that remain growth arrested for long periods post-exposure (5, 19). Phenotypic resistant cell stages can be environmentally induced or stochastically switched. This switch is reversible, showcased by their progeny being equally susceptible as the initial susceptible parental population from which they derived (5, 20). The heterogeneity in phenotypic characteristics illustrates the diversity of phenotypic resistance mechanisms, that is seconded by the diversity in biochemical and molecular phenotypic resistance mechanisms (15, 21–26).

Most investigations on antibiotic action have been performed on batch cultures with large bacterial populations since the clinical aim is to eradicate the pathogenic bacteria population. However, differentiating between survival and growth response, and quantifying heterogeneity among cell responses needs single-cell data (4, 5, 14, 27) [but see also (15, 16, 28)]. In addition, most of these investigations have been made with a single-drug exposure. Yet, pharmacokinetics impose fluctuating levels of antibiotics at sites of infections—when standard oral administration of drugs with 24-h intervals are given (29).

Here we quantify fractions and characteristics of phenotypic resistant cells that arise from an exponentially growing susceptible parental population under recurrent exposure to inhibitory or sub-inhibitory concentrations of ampicillin, a β-lactam antibiotic. We investigate how phenotypic resistant cells differ in their single-cell characteristics from susceptible cells by studying ampicillin-triggered changes in their

phenotypes. We hypothesized that cell mortality would follow distinct patterns during exposure to ampicillin: first, the highest mortality would occur within the initial 1.5 h of exposure, consistent with known pharmacodynamics. Second, mortality rates would increase around the MIC (4 µg/mL) and remain elevated for supra-MIC concentrations. Third, mortality rates would decrease to pre-exposure levels during non-exposure periods but rise again during subsequent drug exposure periods. These subsequent risings should not reach the levels observed during the initial exposure period as fractions of cells should have switch to a phenotypically resistant state.

We further hypothesized for growth characteristics, that antibiotic exposure surviving cells—phenotypic resistant cells—would be cells that showed slow- or no-growth prior to the first (and subsequent) exposure period to a β-lactam, as ampicillin, because ampicillin disrupts cell wall synthesis, killing fast-growing cells to high degrees (4). Such predetermined growth expectations further imply that few cells will stochastically switch to phenotypic resistant stages or switch by environmental induction (5, 30). To gain a wider understanding of phenotypic resistant cell characteristics, we compared the revealed single-cell patterns with bacterial populations mimicking a similar antibiotic exposure regime.

## MATERIALS AND METHODS

We conducted four set of experiments, three of them with three recurring ampicillin exposure periods of 90 min; exposure periods started 60 min, 330 min, and 1,080 min past the onset of the experiments (see also Fig. 2 to 5 and 7 gray vertical bars). Two of these sets collected single-cell data using a mother machine microfluidic device and time-lapse microscopy (Fig. 1; Movie S1) (31–33) and one set collected cell culture level data, that is, CFU counts (details in the supplemental methods and Fig. S5). Cells were derived of an exponentially growing susceptible isoclonal *E. coli* strain K12-MG1655, ATCC 700926TM, population. Exposure was at one of eight different ampicillin concentrations ranging from sub-MIC to supra-MIC levels [0–128 µg/mL; MIC = 4 µg/mL, minimum bactericidal concentration ~16 µg/mL; with 90 min ampicillin exposure periods, and 3-h lasting (first) and 11-h lasting (second) recovery periods; details in the supplemental methods]. Single-cell death in the microfluidic device was determined using propidium iodide. The fourth experiment, which was at the cell culture level, continuously exposed cells to antibiotics and served as a control experiment. All experiments, single-celled and cell culture experiments, were done in supplemented minimal media (M9 +glucose + casamino acids), and all cultures were derived from the same *E. coli* strain, that has a replication time of ~25–30 min in M9 medium at 37°C (details in the supplemental methods). We chose this strain as it is well adapted and explored and has been used in previous work on phenotypic resistance (17, 22, 30, 34). More details on bacterial cell culturing, microchip casting and mounting, microchip loading, life cell imaging, image analysis, and experimental treatment are provided in the supplemental methods. We remain brief in our description here as most methods follow previously published protocols (31, 33).

The resulting data were subsequently analyzed in program R (35) using general linear, generalized linear, and non-linear (Generalized Additive Models, GAM) models. Statistical testing was mainly done by model comparison based on information criteria (AIC) (36). We considered an ΔAIC >2 as better statistical support between competing models. Survival analyses (Kaplan-Meier Fig. 2a) are computed with R package survival (survfit function) and compared among them with a Cox proportional hazard model (coxph) and post hoc testing was done by pairwise comparison (pairwise_survdiff). Probability of death curves (Fig. 2b and 5b), division rate curves (Fig. 4a), and size curves (Fig. 4b) were plotted with ggplot2 package and a loess smoothing (geometric_smooth function; package ggplot2), growth curves were fitted with loess smoothed GAM models (Fig. 4c), competing GAM models (Table 1) were fitted with the restricted maximum likelihood method (REML). Details on the statistical testing, exclusion of extreme values, and model

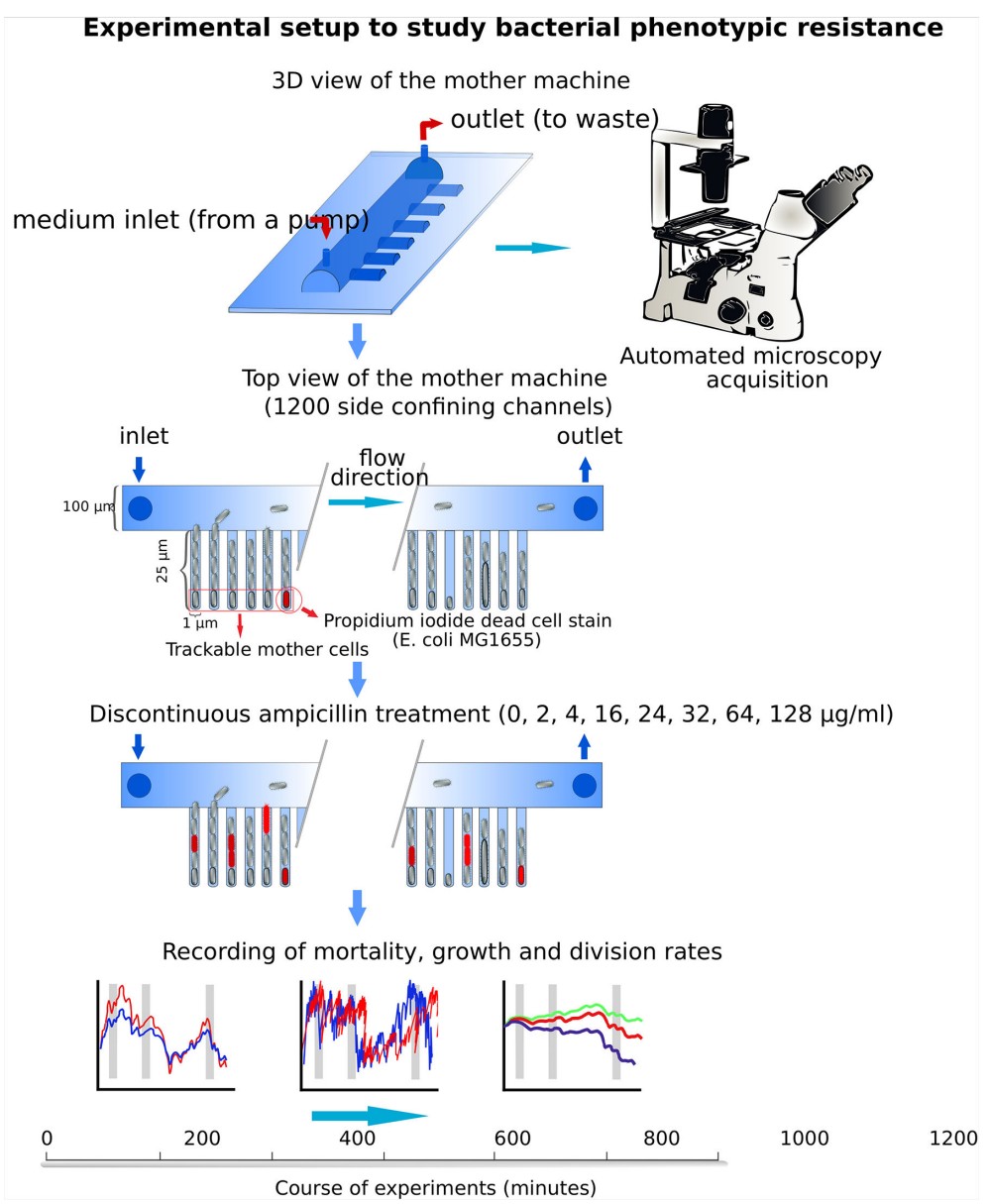

**FIG 1** Experimental setup for studying phenotypic resistance at the single-cell level of *E. coli* MG1655 to ampicillin using the mother machine microfluidic system.

specifics are given in the supplemental methods, which also include details on comparisons of growth differences before death (GLMs; post hoc Tukey test using glht function of mutlcomp package) and testing for the robustness of growth rates (Fig. 6; GAM) (see also the supplemental material, Stats Tables).

For the cell culture experiments, to collect the number of CFU data and the cell culture density data ($OD_{600}$), we initially seeded these experiments using cells from an exponentially growing culture. Each experiment had eight replicates, the experiments were done in 96-well plates, and cells grew at 37°C and vigorous shaking in their respective well (Fig. S5). Recurrent exposure to different antibiotic concentrations was achieved through serial dilution, centrifuging, removing supernatants, resuspending, and culturing cells, followed by serial dilution and plating on agar plates and CFU were counted after 16 h of incubation; more details are provided in the supplemental methods.

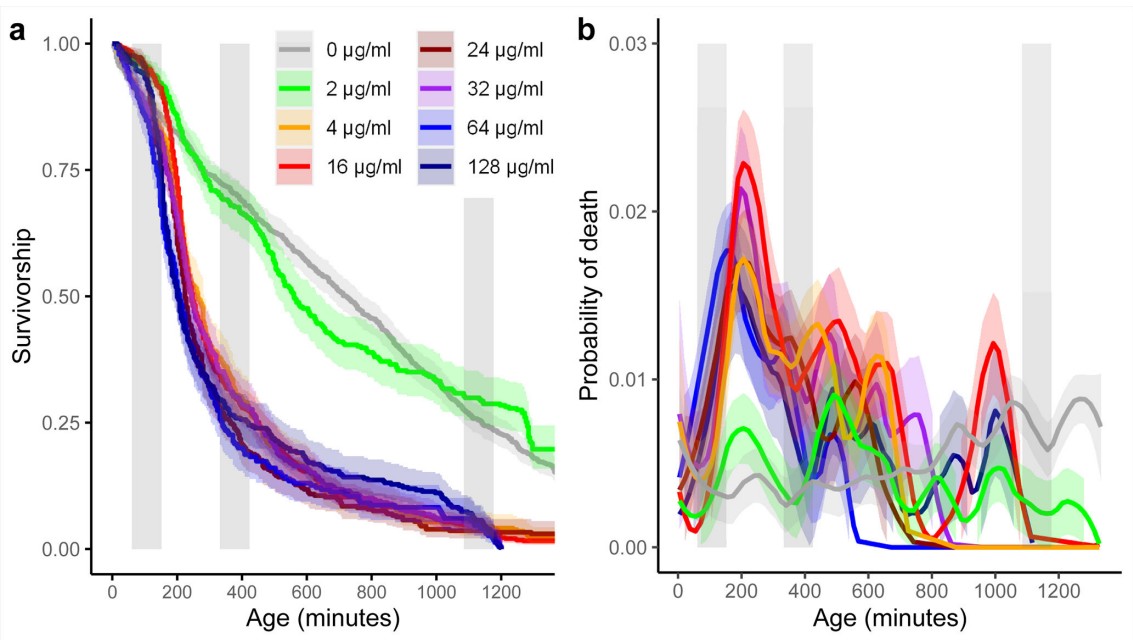

**FIG 2** Survivorship curves (a) and probability of death, that is, a force of mortality (b) of cells exposed to different levels of ampicillin across the duration of the experiment (MIC 4 µg/mL). The gray curves are control group cells that were not exposed to any antibiotics. The gray vertical bars mark the exposure period to the antibiotic. Kaplan-Meier survivorship curves (a) are shown with 95% CI. Probability of death curves (b) is loess smoothed with 95% CI. Statistical testing is shown in Table S1A in the supplemental material. Note, the two panels (a & b) are built from the same single-cell mortality data (n number of cells: 0 µg/mL = 898; 2 µg/mL = 334; 4 µg/mL = 337; 16 µg/mL = 669; 24 µg/mL = 330; 32 µg/mL = 347; 64 µg/mL = 230; 128 µg/mL = 351).

## RESULTS

### Mortality of susceptible and phenotypic resistant cells

As expected, the probability of cell death, that is, the force of mortality, increased after the first exposure of the susceptible parental population to ampicillin, when exposed to MIC (4 µg/mL) or supra-MIC concentrations. Below MIC, no difference in survival was detected (Fig. 2b; Movie S1). When exposed to MIC or higher concentrations, we considered the ~33% of cells that survived the first ampicillin exposure period as phenotypic resistant cells and the ~66% of cells dying before the second exposure period as susceptible cells (Fig. 2a and 3; Movie S1). The survivorship patterns for the above MIC concentrations (Fig. 2a) somewhat resemble a biphasic killing curve as frequently observed at the cell culture level of heterogeneous phenotypic resistant

**TABLE 1** Model comparison among competing GAM models on division rate, size, or growth rate and age and antibiotic concentration[a]

|  |  | DF | AIC |
|---|---|---|---|
| **Division rate** | **S(Age) + Concentr.** | **15.8** | **455,232** |
|  | S(Age) | 8.4 | 487,667 |
|  | Concentr. | 8.0 | 455,263 |
| **Size** | **S(Age) + Concentr.** | **16.8** | **2,332,990** |
|  | S(Age) | 10.8 | 2,340,083 |
|  | Concentr. | 8.0 | 2,337,437 |
| **Growth rate** | **S(Age) + Concentr.** | **17.9** | **−1,033,384** |
|  | S(Age) | 10.8 | −1,016,087 |
|  | Concentr. | 9.0 | −1,029,407 |

[a]For each of the three response variables (division rate, size, and growth rate), we compared three competing models that had two explanatory variables (age, i.e. time since onset of experiment, and antibiotic concentration). The best supported of these three models is highlighted in bold. Division rate is modeled with a binomial error structure, and size and growth are modeled with a Gamma-distributed error structure.

(susceptible and persister) cells. We caution here and extend on it in the discussion, that direct comparison among cell culture level killing curves and single-cell survival can only be made in a crude way. Mortality peaked 30 to 60 min after the first 90-min antibiotic exposure period (Fig. 2b), which indicates a lag time in the effect of killing of 2 to 2.5 h after the cells first experienced any antibiotic (see also Fig. 3). Survival patterns were similar at, or above, the MIC of 4 µg/mL (Fig. 2a, Likelihood ratio test 660.8, $P < 0.0001$, $n = 3496$; for details on Cox proportional hazards and post hoc testing, see Table S1A). The subsequent 90-min exposure period triggered a less pronounced mortality force compared to the first exposure period with a similar lag time of about 30 to 90 min after the second antibiotic exposure (Fig. 2b). Such reduced sensitivity to recurrent antibiotic exposure was expected, as the first exposure period left phenotypic resistant cells as survivors; also recall that the same cells are tracked throughout all the experiment. Note, toward the third exposure period, cell numbers became small (~20–30 cells per

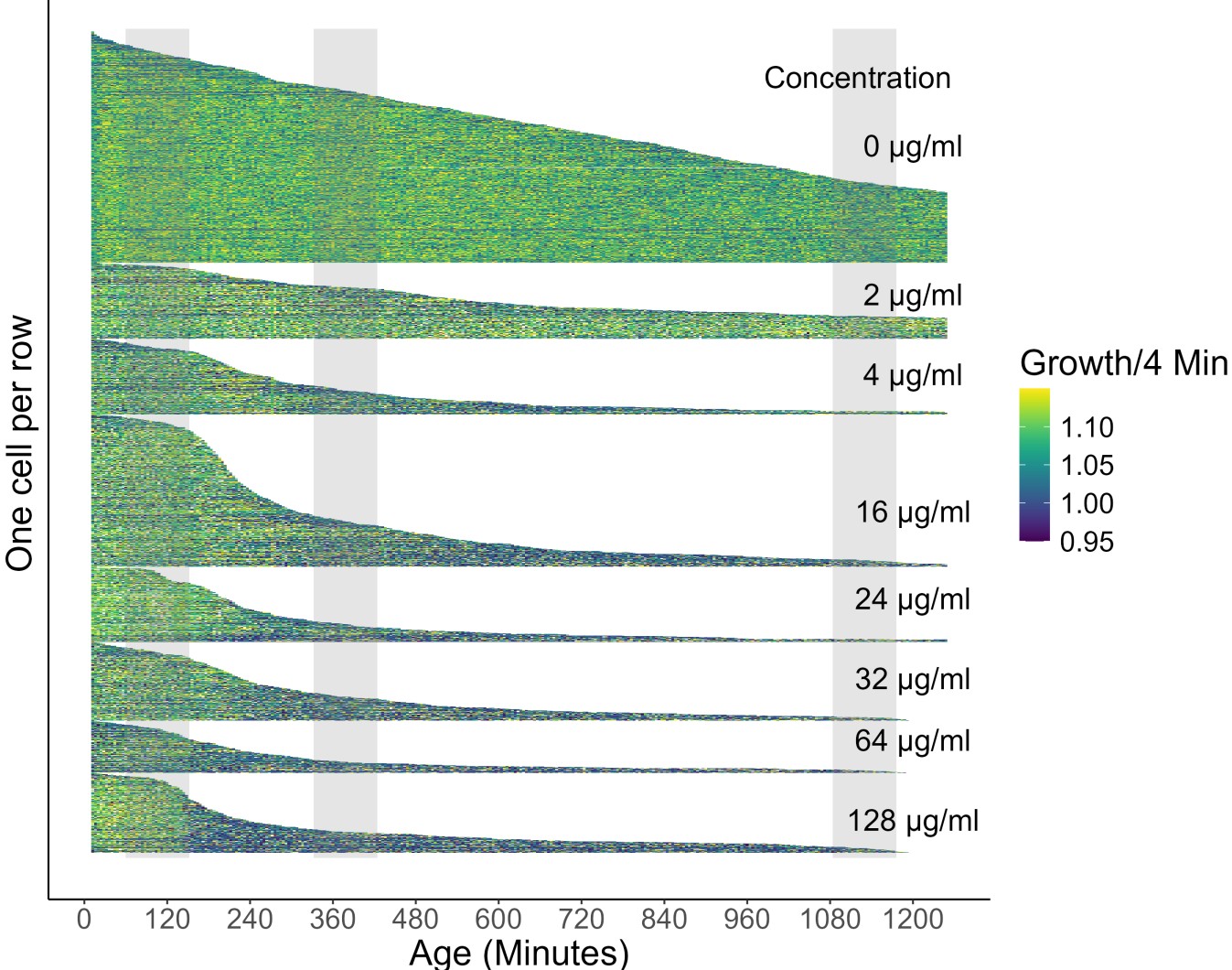

**FIG 3** Single-cell growth dynamics across age for 3615 cells. Each row illustrates the growth dynamics of one cell (color gradient: high growth yellow, shrinkage blue). The length of the rows highlights longevity of each cell, with various cells being right censored at the end of the experiment. Cells are sorted by longevity within their group of exposure to the respective ampicillin concentration. Note that some cells have been washed out of their growth wells while still alive. These cells have been considered as right censored in the other analyses but not specifically marked in this Figure. The gray vertical bars mark the exposure period to the antibiotic. To increase the color contrast of the growth rates, we binned all growth >1.15 into the yellow category and growth rates <0.95 into the dark blue category. N number of cells: 0 µg/mL = 1,017; 2 µg/mL = 334; 4 µg/mL = 337; 16 µg/mL = 669; 24 µg/mL = 330; 32 µg/mL = 347; 64 µg/mL = 230; 128 µg/mL = 351.

concentration) and estimates of mortality rates less reliable (Fig. 3). For the control, 0 µg/mL, and the 2 µg/mL concentration, cells died, as expected, at rates in agreement with experiments exploring aging patterns of single bacteria cells without antibiotic influence, but otherwise similar conditions (supplemented M9, 37°C, high energy light exposure for fluorescent imaging of PI) (31, 33).

## Division rate, size, and growth responses

In contrast to the control group of no antibiotic or sub-MIC exposure, cells that had been exposed to MIC or supra-MIC antibiotic concentrations reduced their division rate after the first exposure period, changing from ~2 to ~1 division per hour (Fig. 4a; Table 1). Models that considered the age (time since the onset of the experiment) and the antibiotic concentration were better supported than models that did not differentiate among age patterns or concentrations (Table 1). Most of the reduction in division rate was reached ~150–200 min after the onset of the first exposure (300–360 min after the onset of experiment). The division rates tended to decline after the second exposure (Fig. 4a). A small decline in division rates might also arise due to senescence in growth and not only antibiotic exposure (33).

Average cell size remained approximately constant within a given antibiotic concentration. Although we found a surprising amount of variation in size among cells exposed to different levels of antibiotics (Fig. 4b; Table 1); there was no relationship between size and the levels of antibiotics. Models that considered the age (time since onset of the experiment) and the antibiotic concentration were better supported than models that did not differentiate among age patterns or concentrations (Table 1), highlighting the variation in size patterns across the experiments and among concentrations.

Cell growth rates responded as predicted by declining during and after the first antibiotic exposure for cells exposed to the MIC or above. We highlight here that growth rates are only approximately halved (Fig. 4c; Table 1), which contrasts our predictions based on previous findings where phenotypic resistant cells were growth arrested or near growth arrested (5) (but see 8, 10, 12–17). Again, both age and antibiotic concentration influenced growth response as illustrated by the best supported models that included both age and concentration as explanatory factors (Table 1). The growth reduction started about 30 min after the onset of antibiotic exposure, which indicates that the growth reduction is an induced response with a minimum lag time that corresponds to approximately the doubling rate (division time) of the parental population. This lag time is also visible when the single-cell data are visualized for each cell

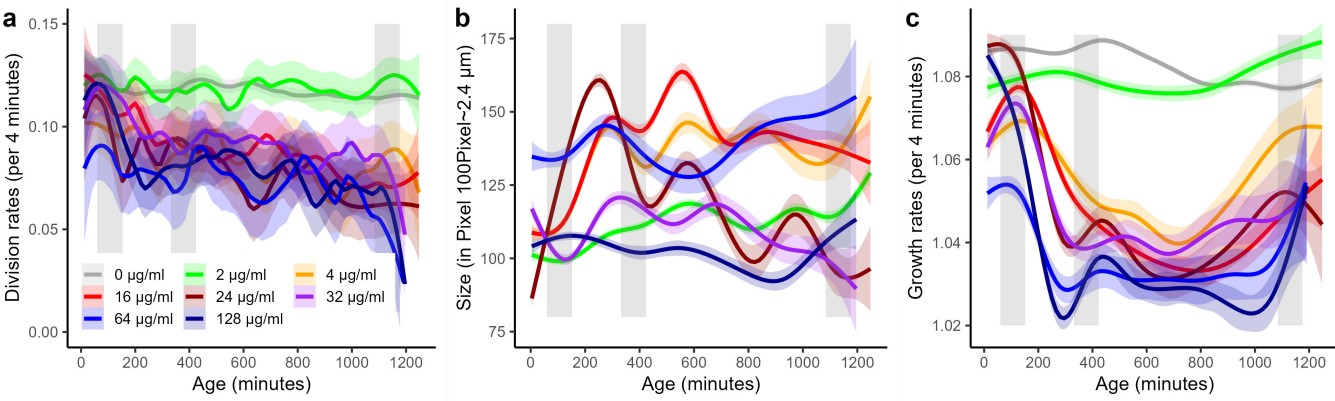

**FIG 4** Cell division rates (a), cell size (in pixels with 100 pixels ~2.4 µm) (b), and growth rates (c) of cells exposed to different levels of antibiotics across the duration of the experiment (MIC 4 µg/mL). The gray vertical bars mark the exposure period to the antibiotics. Division rate (a), size (b), and growth rate (c) curves are with 95% CI. N number of cells: 0 µg/mL = 1,017; 2 µg/mL = 334; 4 µg/mL = 337; 16 µg/mL = 669; 24 µg/mL = 330; 32 µg/mL = 347; 64 µg/mL = 230; 128 µg/mL = 351. Note, for the size data (b) there is no 0 µg/mL data available. Model assumptions are also evaluated in Fig. S1.

by increased frequency of bluish colors after the first exposure period (leftmost vertical grey bar Fig. 3). The lowered growth rates stabilized about 4–5 h after the onset of the initial exposure period (Fig. 4c). Substantial variance in lag time among individual cells' growth response exists (Fig. 3), which also becomes apparent by the gradual decline in average growth rate after antibiotic exposure in Fig. 4c. Induced growth reductions depended partly on antibiotic concentrations with higher levels of antibiotics leading to lower growth rates (Fig. 4c). However, even with the highest antibiotic concentration, the growth rates did not drop below 1.02–1.03/4 min, that is, 2%–3% cell elongation within 4 min (~2.3–1.5 h division time). Furthermore, growth did not resume to prior exposure levels after the release of exposure to the antibiotic.

## Growth-specific response

The growth rates as discussed above (Fig. 4c) are average growth rates at a given point in time, but cells with high growth rates might still suffer higher mortality as expected from a β-lactam antibiotic and as previously described for phenotypic resistant cells (20). To evaluate the possibility of such differential mortality based on cell growth, we combined all cells exposed to MIC or supra-MIC for analysis. We then separated them into three growth categories: cells that did not grow or shrank (growth-arrested cells with growth rates < 1.0), cells that grew slowly (growth rates 1.0–1.1), and cells that grew quickly (growth rates > 1.1) (Fig. 5a). Note that cells can switch dynamically among these categories throughout the experiment (Fig. 3). Furthermore, note that in these analyses mortality and survival are associated only with the current growth rate of a cell and not a cell's growth rate averaged across time.

As expected, fast-growing cells showed increased probabilities of death upon exposure to the β-lactam ampicillin, but contrasting to our expectations, the probability of death of growth arrested cells increased substantially upon exposure to antibiotics, whereas the probability of death of slow-growing cells did not change (Fig. 5b). These results are supported by statistical model comparison, where models including the growth category explained the age at death better than those that did not consider the current growth rate (Table S1B). Shrinking and fast-growing cells had similar mortality, while slow-growing cells differed in their mortality rates. These findings are supported by model comparison for which models did not significantly differ in explaining age at death when only fast-growing and shrinking cells were compared,

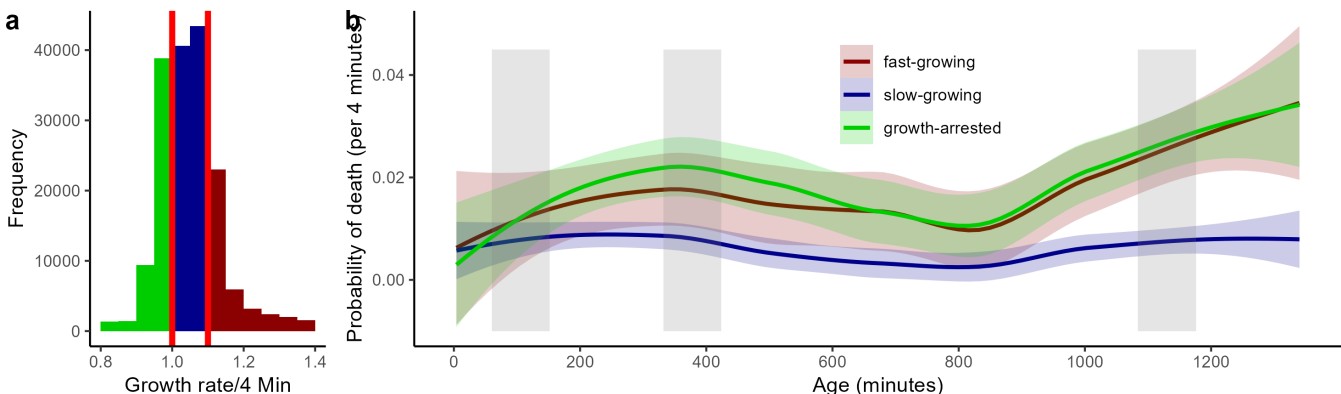

FIG 5  Cell growth rate distribution is divided into three categories: growth arrested (shrinking or non-growing; green), intermediate growth (blue), and fast growth (red) (a), and probability of death for these three groups of growth-arrested, intermediate, and fast-growing cells across the duration of the experiment (b). Confidence intervals (95% CI) are shown in shading in panel (b). The gray vertical bars in panel (b) mark the exposure period to the antibiotics. Note that cells can switch among growth categories throughout the experiment, that is, cells are categorized according to their current cell growth rate. For this analysis, we have combined cells across levels of antibiotic exposure ≥4 µg/mL and excluded all cells exposed to 2 µg/mL. A statistical model comparison showed that mortality differed substantially among the three growth category groups: ΔAIC 25.8 comparing a growth category model and a null model, see also the supplemental material. Classical *P*-value post hoc analyses showed that fast-growing and growth-arrested groups did not differ from each other in age at death, but the intermediate group did.

while models separating slow-growing cells from fast-growing and shrinking cells were better supported in predicting age at death (Table S1C).

To verify the robustness of our finding that mortality is dependent on growth rates, and to broaden our understanding of how growth influences mortality beyond a cell's current growth rate (Fig. 5b), we did three comparisons between average growth rates of cells dying during specific periods in time. First, we compared the average growth rate (averaged across time) of cells that survived the first exposure period to those that died before the first exposure period. In doing so, we evaluated whether phenotypic resistant cells (surviving cells) showed reduced growth rates before their first exposure to antibiotics, an expectation we had based on previously described characteristics of phenotypic resistant cells being growth arrested or near growth arrested before being exposed to antibiotics. Second, we compared the average growth rate of cells dying during or shortly after the first exposure period, which we considered susceptible—antibiotic sensitive—cells, to cells surviving past the first peak of mortality (phenotypic resistant cells). Third, we compared similar cells that died between the first peak of mortality and the second peak of mortality to those that survived past the second peak of mortality (Table 2). In addition, to evaluate growth behavior before the first exposure, we compared growth rates prior to the first exposure of antibiotics among groups of cells that died before the first exposure, that died during the first exposure, that died between the first and second exposure periods, or those that died after the second exposure period (Fig. S2, Table S1E).

Before the first exposure, cells that died had lower average growth rates compared to those that survived past the first exposure, showing that without selection of antibiotics slower growing cells show higher mortality rates. This pattern of increased mortality becomes also partly apparent for the single cell data (Fig. 3, best seen by focusing on 0 µg/mL group) where shorter living cells are showing lower growth (shorter rows are bluish-dominated rows). Cells that died during the first exposure or shortly after (sensitive cells) had slightly higher growth rates than those that survived past this first increased selection period (phenotypic resistant cells). This pattern of growth-dependent selection remained for the next period of increased selection after the second antibiotic exposure. These findings support our initial prediction that faster-growing cells would show higher mortality under antibiotic exposure. However, note that the effect sizes (growth differences) among surviving and dying cells are small, even though they are significantly different. Also, when growth evaluation is limited to the first hour of the experiment (before the first exposure period) and groups of cells that died at different ages (age at death categories) are compared, these groups did not differ in obvious ways, at least visually no clear pattern appears (Fig. S2). Statistically models that accounted for the interaction among age at death categories and antibiotic concentration were still best supported, suggesting some variability among age at death groups and later exposure to concentrations (Table S1E). If phenotypic resistant cells had been already

**TABLE 2** Differences in average growth rates of cells surviving past or dying during or after specific points in time[a]

| Period | 0–60 min | 60–328 min | 328–600 min | +600 min |
|---|---|---|---|---|
| Growth rate of cells dying during period (±Stdev) | 1.047 (±0.039) | 1.067 (±0.032) Susceptible cells | 1.057 (±0.030) | 1.045 (±0.027) |
| Growth rate of cells surviving period (±Stdev) | 1.072 (0.0412) | 1.060 (±0.035) Phenotypic resistant cells | 1.050 (±0.028) | |
| AIC growth during/after | **−8,020.8**[b] | **−8,816.3** | **−3,170.8** | |
| AIC null model | −7,985.4 | −8,772.0 | −3,161.4 | |
| ΔAIC | 35.4 | 44.3 | 9.4 | |

[a]Model comparison among a null model (intercept only model) and a model with the two groups of cells dying during a particular period or surviving that period as the explanatory variable and the mean growth rate (per 4 min) as the response variable. Three sets of models were fit, one for each period (0 min–60 min, 60 min–328 min, and 328 min–600 min). A Gaussian error structure was used to fit these models.
[b]The best supported models based on AIC model comparison (see also ΔAIC) are highlighted in bold.

in a growth-reduced state prior to antibiotic exposure, as frequently reported for other phenotypic persistent cells, a clear reduction in growth rate among the age at death categorized groups would be expected. Combined with the findings shown in Fig. 5, we interpret our findings on growth-specific mortality under antibiotic exposure as not primarily impacting fast-growing cells, and lacking growth differences before antibiotic exposure among susceptible and phenotypic resistant cells.

## Robustness of growth rates within cells

In Fig. 4c, we investigated the average growth rate at a given time point. As cells can dynamically alter growth rates during their course of life (Fig. 3), we correlated a cell's current growth rate to its growth rate in the future, which provides a measure of the robustness of growth. The overall correlation between an individual's cell growth rate at a given time t and its growth rate in the future [time $t + 1$ (4 min later)] or later time points [time $t + 3$ (12 min later); time $t + 5$ (20 min later)] was weak (Fig. 6; Table 3). One can interpret this as growth rates being a little robust. A cell's current growth rate does not well predict its future growth rate. However, there was an exception for cells with slow growth rates (1.0–1.1). Cells growing in that range (Fig. 6, x-axis range 1.0–1.1) showed a strong correlation between current and future growth, illustrating robust constant growth for these slow-growing cells. Growth-arrested cells (growth rate <1 at time t) tended to increase in their growth rates at future points in time, while fast-growing cells (growth rates > 1.1 at time t) tended to decrease their growth rates (Fig. 6). These patterns of growth robustness of slow-growing cells were only slightly tightened by antibiotic exposure, as evidenced when growth correlations are compared before the first antibiotic exposure period with those after the antibiotic exposure period (Fig. S3, Table S1F). Overall, we observed substantial temporal dynamics in growth rates with less robust growing cells (growth-arrested and fast-growing cells) potentially being more affected by the concentration of antibiotics. Individual cell variance in growth prior to antibiotic exposure was not a reliable predictor to identify phenotypic resistant cells, as it did not correlate well with subsequent survival (Fig. S4). It remains that the growth rate responses to different concentrations of antibiotics shown in Fig. 4c are likely influenced by the fastest-growing and growth-arrested cells.

## Population-level responses

Much research investigating phenotypic resistance has been done in cell cultures, that is, at the population level for which averages are estimated. Such averaged estimation makes it challenging to partition mortality, division, and cell growth responses, characteristics that need single-cell level evaluation, as focused on in the sections above.

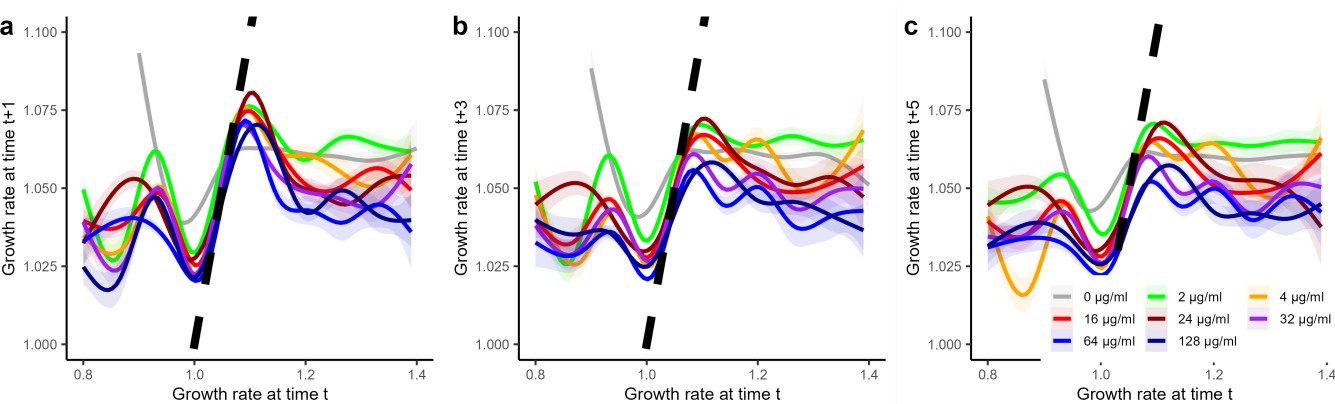

**FIG 6** Correlation in cell growth rates at a given time t, compared to growth rates at time t+1 (4 min later) (a), t+3 (12 min later) (b), and t+5 (20 min later) (c) separated into populations recurrently exposed to different levels of antibiotics (as described in the main set of experiments). All growth rate correlations are fitted with a GAM with 95% CI. The dashed black line illustrates the direct correlation among time points.

**TABLE 3** Model comparison among competing GAM models, for correlation of current growth rates to growth rates at future points in time[a]

|  | Model | DF | AIC |
|---|---|---|---|
| Growth t+1 | **Concentr. + S(Growth by Concentr)** | **78.6** | **−1,005,247** |
|  | Concentr. + S(Growth) | 17.9 | −1,001,617 |
|  | S(Growth) | 11.0 | −992,655 |
|  | Concentr. | 9.0 | −981,293 |
| Growth t+3 | **Concentr. + S(Growth by Concentr)** | **77.7** | **−976,924** |
|  | Concentr. + S(Growth) | 17.9 | −974,444 |
|  | S(Growth) | 10.9 | −967,423.8 |
|  | Concentr. | 9.0 | −953,080 |
| Growth t+5 | **Concentr. + S(Growth by Concentr)** | **77.9** | **−959,919** |
|  | Concentr. + S(Growth) | 17.9 | −956,777 |
|  | S(Growth) | 11.0 | −949,452 |
|  | Concentr. | 9.0 | −938,881 |

[a]Best supported model for each response variable (Growth t+1, t+3, and t+5) is highlighted in bold. GAM models are fitted with the restricted maximum likelihood method, a smoothing parameter for growth rate at time t with a shrinkage version of a cubic regression spline as smoothing parameter (bs = cs) and/or the concentration, including a model that accounts for the interaction of the two explanatory variables, growth rate and concentration of antibiotics. Growth rates are fitted with a Gaussian error structure and an inverse link function.

Furthermore, density dependence and resource depletion need consideration at the cell culture level but are controlled for by the microfluidics setup at the single-cell level. Scaling and comparing single-cell characteristics to population-level patterns can therefore be challenging. In the following section, we report on "control" experiments done in cell cultures. For these cell culture experiments, we took two approaches: CFUs were counted before and after each exposure period to antibiotics (Fig. S5) to mimic the recurrent application of antibiotics as in the single-cell experiments, or we evaluated cell densities (OD) under constant antibiotic concentrations to provide comparison to frequently explored longer exposure to antibiotics of other studies. Cell cultures exposed to sub-MIC concentrations showed similar qualitative dynamics in terms of the numbers of CFUs when recurrently exposed to antibiotics (Fig. 7a; Table S1G). In the single-cell experiments, we did not see much-graded response in mortality for sub-MIC concentration (Fig. 2), but for the cell culture experiments, we find the higher the antibiotic concentration the stronger the reduction in CFUs (Fig. 7a; Table S1H). Relapse in OD after relieving exposure to antibiotics was highest for intermediate antibiotic concentrations (8–32 µg/mL). These patterns might arise due to resource limitations that limit growth (i.e., the cultures approached the stationary phase) during the long recovery period after the second exposure. Such growth limitation seems not to occur when previously exposed to 64 and 128 µg/mL. The decline in CFU during recurrent exposure shows the continued susceptibility of the cells after episodes of heightened mortality due to antibiotic exposure. When populations were permanently exposed to antibiotics (Fig. 7b), they, as expected, initially increased in density before densities were reduced to very low levels for all cultures exposed to MIC or supra-MIC concentrations. The susceptibility of the initial population to antibiotics and no occurrence of dominating resistance mutations during the experiment aligns with these findings and support our claim that genetic resistance mutations should not play any role in the single-cell findings.

## DISCUSSION

We report on the phenotypic characteristics of single cells in *E. coli* in response to recurring exposure to different concentrations of a β-lactam antibiotic, ampicillin. Our findings suggest an additional diversity of modes of phenotypic resistance, given that we show a previously undescribed combination of characteristics. These characteristics might foster genetically fixed resistance evolution as mutational opportunities are widened and periods that allow for horizontal gene transfer of resistance genes are prolonged. Under our experimental conditions, a phenotypic resistant cell holds about

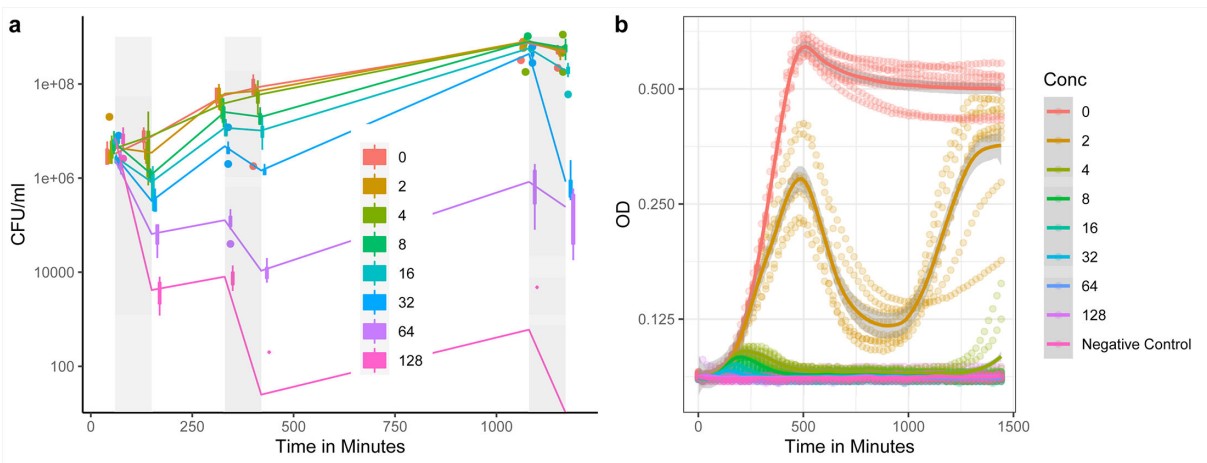

**FIG 7** Colony forming units (CFU) (a) and optical density (OD$_{600}$) (b) of cell cultures exposed to different levels of antibiotics across the duration of the experiment (a) or constant exposure to antibiotics throughout the experiment (b). The gray vertical bars (a) mark the exposure period to the antibiotics. CFUs (a) are plotted as boxplots with a line connecting the mean number of CFU per time when CFU were counted. Single data points are outliers. Note that a jitter is added (a) for better visibility. For the OD curves (b), the dotted lines show the different replicates. For the negative control (b) is no bacteria added to the medium. CFU (a) and OD (b) are plotted on a log scale.

a four times higher probability of evolving genetic resistance mutations compared to a susceptible one (Table S1D). This estimate is based on the expected number of divisions the phenotypic resistant cells, compared to the susceptible cells, undergo during the experiment. Since mutations mainly occur when cells replicate their DNA, a process that happens at each division, the number of divisions should scale with the opportunity and probability for mutations to arise. Similar approaches have been applied in other studies evaluating the evolutionary contribution of phenotypic resistance toward genetically fixed resistance (10). Estimates only based on a number of divisions do not consider factors such as heightened mutation rates of phenotypic resistant cells compared to susceptible cells, or heightened mutation rates under antibiotic exposure, including ampicillin (37, 38). In addition, the contributions of phenotypic resistant cells toward genetic resistance evolution depend on conditions, phenotypic resistance characteristics, and which cell subgroup is defined as phenotypic resistant (10). Studies showed that fluctuating exposure to antimicrobials increases bacterial evolvability (39, 40). Our estimate can only be seen as a crude quantification, but our findings contribute to the accumulating evidence that phenotypic resistance serves as a stepping stone to the evolution of genetic resistance, calling for greater awareness of anti-phenotypic resistance strategies to avoid genetic resistance evolution (37, 41–47).

The phenotypic single-cell characteristics we describe differ from previously described characteristics, which indicates novel underlying mechanisms of phenotypic resistance. Prior to antibiotic exposure, it could not be predetermined by growth rates whether cells would be phenotypically resistant, which aligns with studies that illustrated that phenotypically resistant cells arise from the exponential growing populations without special growth characteristics (10, 14, 16). These more recent findings contrast with classical studies where growth-arrested cells were subsequently identified as phenotypic resistant cells (4). The emergence of phenotypic resistant cells out of an exponentially growing population as we show suggests that the evolution, or the maintenance, of phenotypic resistance, is not conditional on nutrient-limiting conditions as has previously been suggested to be the favorable evolutionary route (48). The lack of distinct growth difference between susceptible and phenotypic resistant cells prior to drug exposure also suggests that phenotypic resistant states were triggered by the antibiotic and were not stochastically switched to as rarely described (4). The lag phase to achieve reduced growth also suggests a triggered response, and the variance in time to switch to a phenotypic resistant—growth reduced—state indicates that phenotypic

resistant cells differ in how long it takes to transition to the characteristics of a resistant state (49, 50).

Although the molecular mechanism of how *E. coli* becomes less susceptible to ampicillin upon preexposure is beyond the scope of this article, it is known that ampicillin can induce several mechanisms that limit antibiotic damage and increase survival. For instance, sublethal treatment with ampicillin in *E. coli* can activate the RpoS-mediated general stress response (38), which limits the growth rate and hence general susceptibility (51). In addition, ampicillin also induces or activates the MarA-regulated multidrug AcrAB-TolC efflux pump system, which hinders the efficacy of several antimicrobials (52, 53). One additional consequence of β-lactam exposure in *E. coli* is the activation of the synthesis of colanic acid capsular polysaccharide (54), which, in turn, limits the access of the antibiotic to the cell wall and has been associated with persistence (55). In our microfluidic setup, in comparison with classical experimental procedures, bacteria receive instant *ad libitum* nutrient supply of energy and removal of metabolic waste that might allow cells a quick repair of antibiotic damage and running costly efflux systems such as AcrAB-TolC and sodium-potassium pumps to maintain the osmotic equilibrium (56). In general, the efficacy of antibiotics, including ampicillin, is linked to bacterial respiration rate (51), but we can only speculate on potential mechanisms of how the cells we observe achieve growth under ampicillin. A more detailed molecular investigation would be needed to better understand how the patterns we detect arise.

During antibiotic exposure, previous studies described that most phenotypic resistant cells were growth arrested or nearly growth arrested. Growth arrest is associated with reduced metabolism and respiration, and in combination with general stress responses, these remain the most commonly described mechanism to achieve resistance (5, 14, 16, 51). Our study, surprisingly, contrasts these general patterns of growth arrest, as we find reduced growth rates, but not near-growth-arrest. We find these growth reductions primarily for supra-MIC concentrations, and such reduction quantitatively aligns well with another study that showed responses to two out of seven tested antibiotics when exposed to sub-MIC or MIC concentrations (10). One mechanistic explanation of such non-growth arrested phenotypic resistance is high efflux activity that can lead to the avoidance of accumulation of antibiotics, as described by single-cell studies on fast-growth phenotypes (15). However, such arguments on high efflux activity preventing antibiotic accumulation do not align with the fact that antibiotics in exponentially growing cells usually accumulate within minutes under drug exposure (23). The indication that phenotypic resistance is both antibiotic specific—two out of seven antibiotics triggered reduced growth (10)—and potentially concentration specific, calls for a broader exploration of strains, antibiotics, and concentrations, and deeper exploration of mechanisms that allow for (reduced) growth under antibiotic exposure. Note that the phenotypic characteristics we describe might hold only for our conditions. A broader exploration might also challenge the assumption that phenotypic resistance is a rather general and less antibiotic-specific response compared to the prevalent view that antibiotic-specific mutations cause genetic resistance (5, 8, 57).

Post-antibiotic exposure phenotypic resistant cells often achieve growth resumption to pre-exposure levels within ~3–4 h (time also condition dependent) (3, 5, 13, 16). For the phenotypic resistant cells we describe here, we do not see such growth resumption, and cells stay in their growth-reduced phenotypic resistant state. Other phenotypic resistant cell types, such as sleeper, dormant, or viable but nonculturable cells, remain in their phenotypic resistant growth state for long times, but for them, it is full growth arrest during and after drug exposure. These cells are also characterized by their small cell sizes (19), characteristics we do not find. As mentioned earlier, the different characteristics of phenotypic resistance influence the evolutionary potential of genetic resistance evolution. If we contrast phenotypic resistant cells, that hold the characteristics we describe—in contributing to population growth before, during, and after antibiotic exposure—to "classical" persister cells that show growth arrest prior to

and during drug exposure but regain pre-exposure growth rates after ~4 h post-exposure, the mutation opportunities differ. Under our environmental conditions, phenotypic resistant cells with characteristics we described would have divided ~4–5 times before a classical persister cell would start to divide. This means that for a 1.5-h drug exposure period, each cell holding the characteristics we describe would have produced ~16–32 cells by the time a classical characterized persister cell would start to divide. It would take ~11 h for the classical persister cell (~2 divisions/h) to catch up to the continued slow-growing phenotypic cells (~1 divisions/h) we characterized here. Obviously, conditions vary which affect characteristics and therefore quantitative estimates can at most give an approximate understanding of effects on contributions to genetic resistance, but it becomes clear that evolutionary contributions to genetic resistance vary among phenotypic resistant cells with different characteristics and recurrence of drug exposure.

Quantifying phenotypic resistance contributions to relapsing pathogenic infections can also be challenging as *in vitro* conditions vary among studies. This does not only apply to the type of antibiotics, their concentration, the strain type, the growth media, or the stage the cells are in when exposed (stationary vs exponential) but also to exposure periods. Persister studies often expose susceptible parental populations to 5 h of antibiotics. Subpopulations that build a small fraction of phenotypic persister cells are often in the range of <0.1%, again depending on conditions and persistence mutations (3–5, 15, 16, 28). If we compare to the fraction of cells that survive 90-min ampicillin exposure (Fig. 2a) and extrapolate from there—applying the peak mortality rate of ~0.02/4 min (Fig. 2b) onwards—after 5 h of theoretical exposure ~22% of cells would be expected to be alive, that is, phenotypic resistant. This extrapolated estimate is a little lower than the ~25%–30% survivors we observe 5 h after the onset of the first exposure period (Fig. 2a, see also findings at cell culture for <64 µg/mL; Fig. 7a). Condition differences alone are likely not explaining a 100 times higher fraction of phenotypic resistant cells compared to classical persister cell fractions (3–5, 15, 16, 28), and even the ~2% of perseverant cells arising from exponential phase bacteria that continue to grow at reduced rates under antibiotic exposure, seem to build a 10 times smaller fraction (10). We choose 1.5 h as the ampicillin exposure period, as this time exceeds the pharmacodynamics actions of antibiotics (58). The time to kill a susceptible cell by antibiotics is often being comparable with bacterial generation time (11), which is under our conditions ~30–60 min (Fig. 4a). This time-to-kill action is also revealed by studies that report on reduced mortality rates leading to bi- or multiphasic kill curves caused by phenotypic resistant cells of exponential-phase bacteria after 2 h of exposure (11). The single-cell survival curves we report on (Fig. 2a) resemble those of a biphasic culture level killing curve but with a prolonged time of initiation of the second killing phase after ~4 h. Comparing biphasic pattern between batch culture studies and single-cell studies is challenging, as they employ different measures. For instance, batch culture estimates include daughter cells produced during the killing phase while single-cell data are limited to the initial cells that were loaded at the onset into the microfluidic device. Scaling average division rates of single cells to cell culture growth also suffers from estimation biases (59, 60). Bacterial batch and mother machine cultures exhibit distinct characteristics. The mother machine behaves like a chemostat, but unlike chemostats, confined cells that cease dividing are not washed away from the system. In the mother machine, indefinite exponential growth conditions are sustained by continuously supplying nutrients and removing waste products. When exposed to antibiotics, such as ampicillin, the dynamic environment facilitates continuous proliferation that could alter antibiotic susceptibility dynamics compared to batch cultures, where medium composition slowly changes during the exponential phase. The continuous antibiotic administration might influence characteristics, regardless of the amount of antibiotic bound to its molecular target of living or dead cells. For some infection situations, it is conceivable that the mother machine system better represents the *in vivo* infection dynamic than a batch culture. Thus, comparing single-cell and

cell culture patterns can help to evaluate the robustness of single-cell observations. For instance, and as expected, the MIC marked a survival threshold both in single-cell data and at the growth culture level, and continued sensitivity to antibiotic exposure in mortality (respective CFU) illustrates that the parental population was susceptible and not genetically resistant (Fig. 2 and 7). We also find no graded survival response to different levels of antibiotics at the single-cell level (Fig. 2 and 4a), which supports studies that show insensitivity of fractions of phenotypic resistant cells up to ~100 µg/mL (61), although at the cell-culture level we observed graded responses in CFUs (Fig. 7). We were surprised that slow-growing cells exhibited high survival and robust growth rates under β-lactam exposure compared to growth arrested and fast-growing cells (Fig. 5 and 6) (3, 5, 20, 30). Such slow and robust growth might indicate cell intrinsic homeostasis as a phenotypic resistance characteristic, but such homeostasis would contrast with molecular-level findings that link phenotypic resistance to a perturbed biological network (57).

Our study strengthens that phenotypic resistance has diverse modes of action and that there are various evolutionary routes to phenotypic resistance (5, 6, 12, 16, 57, 62). The diversity of evolutionary paths to phenotypic resistance is illustrated by experimental evolution studies that evolved high phenotypic resistance by repeated exposure to ampicillin and showed that resistance was achieved through increased lag time (50); other experimental evolution studies, which exposed stationary phase cells, evolved phenotypic resistance without prolonged lag times (63), and still other experimental evolution studies found again other, lag time independent, mutations (57, 64). All these investigations suggest that there is substantial diversity in the evolutionary path to phenotypic and consequently genetic resistance and that such evolution might be condition dependent. Future investigations should explore varying drug exposure times and varied inter-exposure time periods to better understand how selection and characteristics depend on exposure periods. Prolonged treatment cycles can cause tolerant strains to attain resistance at higher rates (42), while multi-drug treatment can prolong the time to persistence evolution (65). Strains, antibiotic type, and concentrations should be more systematically explored as small differences in experimental settings can alter outcomes (57). A rigorous exploration of phenotypic characteristics, as we have done, can provide synergistic insights for tailored molecular exploration. For instance, the cells in our study did not show alterations in cell size, which suggests that the phenotypic resistance we explored might not be related to known resistance mechanisms to β-lactams, where filamentation rate is increased, which, in turn, stops ampicillin from efficiently targeting the cells (14, 64, 66). The diversity and partly contrasting findings on phenotypic resistance characteristics illustrate that we are still on the verge of understanding the role and mechanisms of such non-genetically determined resistance. Nevertheless, our results support the idea of diverse modes toward phenotypic resistance, where there is no pre-determined slow growth, no growth arrest during drug exposure, and no quick resumption of growth. These characteristics could foster the evolution of resistance in providing evolutionary reservoirs and increased mutant supply.

Combined, our results show that the susceptible clonal parental population consists of phenotypically heterogeneous cells, whereby the cells have different capacities to survive exposure to antibiotics, leaving predominantly phenotypic resistant cells after a first exposure period. As for applied implications on medical treatment, we need to be cautious about interpretation, because directly relating *in vitro* to *in vivo* results for antibiotic treatment remains challenging. Still, a quantitative understanding of phenotypic resistance is crucial to better evaluate the risk of developing genetically fixed-resistance and understand persistent pathogen infections such as urinary tract infections (UTIs) (10, 37). For such infection, despite killing many cells upon initial antibiotic treatment, the invading apathogenic *E. coli* (UPEC) can persist for months within the bladder epithelium. This persistence is facilitated by the formation of intracellular bacterial communities, which provide a protected niche for bacterial growth

and evasion of host defenses and lower antibiotic concentrations (67, 68). Anti-phenotypic resistance strategies should be explored within antibiotic treatment to reduce the risk of genetic resistance evolution and subsequent treatment failure including recurrent infections.

## ACKNOWLEDGMENTS

We thank Jens Rolff and Sophie Armitage for their comments and discussions.

This work was supported by the Deutsche Forschungsgemeinschaft (DFG, German Research Foundation), 430170797 and 430174701.

## AUTHOR AFFILIATIONS

[1]Department of Biology, University of Southern Denmark, Odense, Denmark
[2]Biological Institute, Freie Universität Berlin, Berlin, Germany

## AUTHOR ORCIDs

Alexandro Rodríguez-Rojas http://orcid.org/0000-0002-4119-8127
Ulrich K. Steiner http://orcid.org/0000-0002-1778-5989

## FUNDING

| Funder | Grant(s) | Author(s) |
| --- | --- | --- |
| Deutsche Forschungsgemeinschaft (DFG) | 430170797 | Ulrich K. Steiner |
| Deutsche Forschungsgemeinschaft (DFG) | 430174701 | Alexandro Rodríguez-Rojas Ulrich K. Steiner |

## AUTHOR CONTRIBUTIONS

Silvia Kollerová, Conceptualization, Data curation, Formal analysis, Investigation, Methodology, Writing – review and editing | Lionel Jouvet, Data curation, Formal analysis, Methodology, Software, Supervision | Julia Smelková, Formal analysis, Investigation, Visualization | Sara Zunk-Parras, Data curation, Investigation, Methodology, Writing – review and editing | Alexandro Rodríguez-Rojas, Conceptualization, Data curation, Formal analysis, Investigation, Methodology, Project administration, Supervision, Writing – review and editing | Ulrich K. Steiner, Conceptualization, Formal analysis, Funding acquisition, Investigation, Methodology, Project administration, Supervision, Validation, Visualization, Writing – original draft

## ADDITIONAL FILES

The following material is available online.

### Supplemental Material

**Supplemental Material (mSystems00256-24-s0001.pdf).** Additional details on experiments, conditions, and statistical analyses, as well as supplemental tables and figures.
**Legend (mSystems00256-24-s0002.pdf).** Movie S1 legend.
**Movie S1 (mSystems00256-24-s0003.mp4).** Exemplar view of mother machine with recurring exposure to antibiotics.

### Open Peer Review

**PEER REVIEW HISTORY (review-history.pdf).** An accounting of the reviewer comments and feedback.

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
