## [Reviewer comments · mSystems]

Phenotypic resistant single-cell characteristics under recurring ampicillin antibiotic exposure in *Escherichia coli*.

Silvia Kollerová, Lionel Jouvét, Julia Smelková, Alexandro Rodriguez-Rojas, Sara Zunk Parras, and Ulrich Steiner

Corresponding Author(s): Ulrich Steiner, Freie Universitat Berlin

Review Timeline:

Submission Date:	February 22, 2024
Editorial Decision:	March 20, 2024
Revision Received:	April 30, 2024
Accepted:	May 20, 2024

Editor: Li Cui

Reviewer(s): The reviewers have opted to remain anonymous.

Transaction Report:

DOI: <https://doi.org/10.1128/msystems.00256-24>

Re: mSystems00256-24 (Phenotypic resistant single-cell characteristics under recurring ampicillin antibiotic exposure in *Escherichia coli*.)

Dear Dr. Ulrich Karl Steiner:

Thank you for the privilege of reviewing your work. Below you will find instructions from the mSystems editorial office, and the reviewer comments.

Revision Guidelines

Sincerely,
Li Cui
Editor
mSystems

Reviewer #2 (Comments for the Author):

The revised manuscript by Kollerová et al. has improved significantly based on the reviewers' suggestions. However, there are still some areas where the clarity of the writing could be improved.

1. Introduction:

The introduction is too long (4 pages) and should be shortened to focus on the key scientific questions that the study addresses. The detailed description of the experimental setup (Lines 103-133) should be shortened. Additionally, the authors should avoid

presenting hypotheses without providing sufficient justification.

2. Terminology:

The manuscript uses terms related to antimicrobial susceptibility inconsistently. The authors should use precise definitions throughout the text, following established conventions. For example, they should distinguish between phenotypic resistance and tolerance.

3. Discussion:

Some of the conclusions presented in the Discussion section are not supported by the results, such as the claim that a phenotypic resistant cell has a four times higher potential to contribute to the emergence of genetic resistance mutations compared to a susceptible one (SI Table D). The emergence of genetic resistance mutations is influenced by genetic background, growth/metabolism, and other factors, and it is misleading to make such a general statement without considering these factors.

The authors should better connect their findings to existing knowledge by comparing their results with previous studies on exposure times, persistence fraction, and growth patterns.

4. Broader implications:

The authors should discuss the implications of their findings for understanding how phenotypic resistance in bacteria under long-term antibiotic exposure affects genetic evolution and the development of genetic resistance.

What are the broader implications of the study for the treatment of bacterial infections?

Reviewer #4 (Comments for the Author):

In this manuscript, Kollerová et al investigate phenotypic characteristics of thousands of single *E. coli* cells upon repeated exposure to the antibiotic ampicillin. Using time-lapse microscopy and a microfluidic device, they find that a fraction of cells from an isogenic population survives lethal concentrations of ampicillin. Using advanced statistical analyses, the authors explore the relation of phenotypic properties to survival and conclude that cells exhibiting average growth-rate characteristics are more likely to survive ampicillin treatment compared to growth arrested or fast-growing cells. The paper fits the scope of mSystems, it is well written, and the results are presented adequately. Methods and especially data analysis are generally sound. However, I have some concerns listed below.

Specific comments:

1. The terminology 'phenotypic resistance' to describe the observations here is not my preference, however given the multitude of definitions in the field and the fact that the authors explain the terminology in the introduction, this is fine.
2. Could the authors elaborate on why under their microfluidic conditions, they find such high frequencies of phenotypic resistant cells? This is in comparison to other work from the field, but also compared to the data from their cell culture experiments (figure 7), where only 1 in 100 to 1 in 1000 cells survives the first exposure to 128 µg/ml.
3. It seems like cells are only given 1 hour to equilibrate after centrifugation and loading into the mother machine, and then antibiotic treatment is applied. Could this affect the results? For instance, in Figure 4 b and c, it looks like large variation in cell size and growth rate prior to the first exposure to antibiotics is present. How can the different conditions be comparable if such large variation is present before the start of the experiment? Or is this an artifact of how the data is displayed?
4. The units of the growth-rates are not clear to me, and it is also not very clear how they are calculated, i.e. log-transformed or linear length-data? Usually, specific growth-rate with units 1/h or 1/min is calculated. Here typical values would be ~0.7 or 0.01, respectively, for one (size) doubling per hour. Authors give values around 1.08 for untreated control, which I assume would translate to growth of 8 % length within 4 minutes, which would translate to a specific growth-rate of 1.15 1/h, i.e. 1.67 (size) doublings/hour or 0.08 (size) doublings/4 minutes. I would recommend authors to clarify to facilitate understanding. E.g. in line 205 authors refer to this more generally used terminology.
5. L317: The authors bring up that cells which grow less robust exhibit higher mortality. Have the authors looked at whether variability in growth rate per cell prior to antibiotic exposure would be a good predictor for survival?
6. Figure 3a (this seems to be confused with figure 2, see comment 11 below): Why are so many cells dying in the absence of antibiotics? It seems like ~90% of the population is lost over less than 24 hours. Is this actual death or is it related to cells escaping the mother machine channels? Authors mention in L198ff that these death rates are expected, quoting their own work. But how do these high death values relate to e.g. the work by Jun group which introduced the mother machine in 2010 (Wang et al, 2010, Curr. Biol.)? They show an average life span of 100 - 150 generations. I would appreciate if this could be clarified.
7. L352-354: The authors state this as a fact, although this is hypothetical, even though I agree on the potential implication for resistance evolution.
8. L354: It is not clear how the authors arrive at the 4-fold higher potential without referring to the SI.
9. Check use of resistance vs resistant throughout the manuscript, e.g. L25: phenotypic 'resistance cells' to 'resistant cells', also L88
10. L103: arouse is used instead of arose/arise
11. L181-190: Figure references are off. It seems like figure 2 and 3 were confused in the submission system.
12. L187-188: This sentence could require some context, e.g. referring the reader to the discussion section.

13. L382: Authors argue that reduced growth-rate lowers 'general susceptibility' after showing that there is no evidence for this in their data.
14. L385: colanic acid, not colonic
15. Figure 2: Units of Age on x-axis (I assume minutes)
16. Figure 5: Here cells are divided into low, medium and high, whereas in the text they are referred to as growth-arrested, slow-growing and fast-growing (starting from L242).

In this manuscript, Kollerová et al investigate phenotypic characteristics of thousands of single *E. coli* cells upon repeated exposure to the antibiotic ampicillin. Using time-lapse microscopy and a microfluidic device, they find that a fraction of cells from an isogenic population survives lethal concentrations of ampicillin. Using advanced statistical analyses, the authors explore the relation of phenotypic properties to survival and conclude that cells exhibiting average growth-rate characteristics are more likely to survive ampicillin treatment compared to growth arrested or fast-growing cells. The paper fits the scope of mSystems, it is well written, and the results are presented adequately. Methods and especially data analysis are generally sound. However, I have some concerns listed below.

Specific comments:

1. The terminology 'phenotypic resistance' to describe the observations here is not my preference, however given the multitude of definitions in the field and the fact that the authors explain the terminology in the introduction, this is fine.
2. Could the authors elaborate on why under their microfluidic conditions, they find such high frequencies of phenotypic resistant cells? This is in comparison to other work from the field, but also compared to the data from their cell culture experiments (figure 7), where only 1 in 100 to 1 in 1000 cells survives the first exposure to 128 µg/ml.
3. It seems like cells are only given 1 hour to equilibrate after centrifugation and loading into the mother machine, and then antibiotic treatment is applied. Could this affect the results? For instance, in Figure 4 b and c, it looks like large variation in cell size and growth rate prior to the first exposure to antibiotics is present. How can the different conditions be comparable if such large variation is present before the start of the experiment? Or is this an artifact of how the data is displayed?
4. The units of the growth-rates are not clear to me, and it is also not very clear how they are calculated, i.e. log-transformed or linear length-data? Usually, specific growth-rate with units 1/h or 1/min is calculated. Here typical values would be ~0.7 or 0.01, respectively, for one (size) doubling per hour. Authors give values around 1.08 for untreated control, which I assume would translate to growth of 8 % length within 4 minutes, which would translate to a specific growth-rate of 1.15 1/h, i.e. 1.67 (size) doublings/hour or 0.08 (size) doublings/4 minutes. I would recommend authors to clarify to facilitate understanding. E.g. in line 205 authors refer to this more generally used terminology.
5. L317: The authors bring up that cells which grow less robust exhibit higher mortality. Have the authors looked at whether variability in growth rate per cell prior to antibiotic exposure would be a good predictor for survival?
6. Figure 3a (this seems to be confused with figure 2, see comment 11 below): Why are so many cells dying in the absence of antibiotics? It seems like ~90% of the population is lost over less than 24 hours. Is this actual death or is it related to cells escaping the mother machine channels? Authors mention in L198ff that these death rates are expected, quoting their own work. But how do these high death values relate to e.g. the work by Jun group which introduced the mother machine in 2010 (Wang et al, 2010, Curr. Biol.)? They show an average life span of 100 – 150 generations. I would appreciate if this could be clarified.
7. L352-354: The authors state this as a fact, although this is hypothetical, even though I agree on the potential implication for resistance evolution.
8. L354: It is not clear how the authors arrive at the 4-fold higher potential without referring to the SI.

9. Check use of resistance vs resistant throughout the manuscript, e.g. L25: phenotypic 'resistance cells' to 'resistant cells', also L88
10. L103: arouse is used instead of arose/arise
11. L181-190: Figure references are off. It seems like figure 2 and 3 were confused in the submission system.
12. L187-188: This sentence could require some context, e.g. referring the reader to the discussion section.
13. L382: Authors argue that reduced growth-rate lowers 'general susceptibility' after showing that there is no evidence for this in their data.
14. L385: colanic acid, not colonic
15. Figure 2: Units of Age on x-axis (I assume minutes)
16. Figure 5: Here cells are divided into low, medium and high, whereas in the text they are referred to as growth-arrested, slow-growing and fast-growing (starting from L242).

Response to the Reviewers:

Reviewer #2.

We sincerely appreciate your helpful and constructive comments on our manuscript. We have carefully considered your comments and suggestions and are grateful for the opportunity to address them.

1. The introduction is too long (4 pages) and should be shortened to focus on the key scientific questions that the study addresses. The detailed description of the experimental setup (Lines 103-133) should be shortened. Additionally, the authors should avoid presenting hypotheses without providing sufficient justification.

Response: In response to your suggestion, we have shortened the introduction, including condensing the detailed description of the experimental setup as suggested. Additionally, we have ensured that all hypotheses presented are accompanied by sufficient justification (Lines 91- 103), thereby enhancing the overall clarity of the introduction.

2. Terminology: The manuscript uses terms related to antimicrobial susceptibility inconsistently. The authors should use precise definitions throughout the text, following established conventions. For example, they should distinguish between phenotypic resistance and tolerance.

Response: To address this concern, we have carefully reviewed and revised the manuscript to ensure precise and consistent terminology. The first and most important established conventions distinguish between genetically fixed resistance and non-genetic, non-heritable, phenotypic resistance (lines 44-47). Tolerance can be conflicting with this definition as it can be defined to be genetically determined or not. Tolerance is often defined as a population wide response that might, or might not, entail fixed genetic alterations, but heterotolerance can also exist, and persistence, which is non-genetically determined, is conventionally defined as such heterotolerance. In shortening the introduction, we clarify the definitions we use, and avoid terms that might raise ambiguity (Lines 53-70).

3. Discussion. Some of the conclusions presented in the Discussion section are not supported by the results, such as the claim that a phenotypic resistant cell has a four times higher potential to contribute to the emergence of genetic resistance mutations compared to a susceptible one (SI Table D). The emergence of genetic resistance mutations is influenced by genetic background, growth/metabolism, and other factors, and it is misleading to make such a general statement without considering these factors. The authors should better connect their findings to existing knowledge by comparing their results with previous studies on exposure times, persistence fraction, and growth patterns.

Response: We recognize that the claim regarding the potential contribution of phenotypic resistant cells to the emergence of genetic resistance mutations was oversimplified. We have toned down this claim reduced to the probability of the emergence of resistance by mutation based on the number of divisions of phenotypic resistant cells compared to susceptible cells. We also explicitly address and highlight that these estimates are only crude estimates that are condition dependent (Lines 322-340).

We backed our statement with additional references from the literature that illustrate how phenotypic resistance increase the probability of evolution of genetically fixed resistance. We further present and discuss our findings in the light of studies that differ in exposure times, persistence fractions, and growth, all of which are important aspects and contribute to the diversity of phenotypic characteristics observed (Lines 412-431).

4. Broader Implications: The authors should discuss the implications of their findings for understanding how phenotypic resistance in bacteria under long-term antibiotic exposure affects genetic evolution and the development of genetic resistance. What are the broader implications of the study for the treatment of bacterial infections?

Response: We have expanded the discussion to address this aspect in various sections, considering the implications of our research for informing strategies for combating antibiotic resistance and improving treatment outcomes for bacterial infections (e.g. Lines 489-495).

Reviewer #4.

We thank the referee for the thorough evaluation of our manuscript and providing very helpful feedback. We have carefully reviewed each of your comments and suggestions and address them below.

1. The terminology 'phenotypic resistance' to describe the observations here is not my preference, however given the multitude of definitions in the field and the fact that the authors explain the terminology in the introduction, this is fine.

Response: Thank you for your understanding. We fully agree that the current situation on the complexities surrounding terminology in this field is not ideal. As requested by another reviewer, we have explained our usage of the term 'phenotypic resistance' in a shortened introduction to ensure clarity for readers (Lines 44-49 & 69-70).

2. Could the authors elaborate on why under their microfluidic conditions, they find such high frequencies of phenotypic resistant cells? This is in comparison to other work from the field, but also compared to the data from their cell culture experiments (figure 7), where only 1 in 100 to 1 in 1000 cells survives the first exposure to 128 µg/ml.

Response: We very much appreciate this question. The high frequencies of phenotypic resistant cells observed in our microfluidic conditions compared to other studies, including our own cell culture experiments, may be attributed to various factors such as differences in growth conditions, nutrient availability, and stress responses. We have provided some additional explanation on this question and elaborated deeper on the reasons for this discrepancy (Lines 170-173, 347-349, 364-368, 372-381). A much deeper analysis would be needed to understand the details of the underlying mechanisms, which goes beyond this study. We felt at unease to speculate too much on such mechanisms which might likely be manyfold, and we do not see ourselves in a position to exhaustively cover potential causes. Conditions, including growth media, are known to influence susceptibility, so does nutrient availability, all aspects we now mention.

3. It seems like cells are only given 1 hour to equilibrate after centrifugation and loading into the mother machine, and then antibiotic treatment is applied. Could this affect the results? For instance, in Figure 4 b and c, it looks like large variation in cell size and growth rate prior to the first exposure to antibiotics is present. How can the different conditions be comparable if such large variation is present before the start of the experiment? Or is this an artifact of how the data is displayed?

Response: We acknowledge this concern. From previous experiences on the system of our group and other groups, 60 min equilibration had been sufficient for growth stabilization under similar conditions. The GAM models are known to be sensitive to extreme values at the limits of the range

explored, which can explain part of the high variability at the onset (but also the very end of the experiments), other forms of models and visualization could reduce variability at the extremes but would be less efficient in visualizing the core period of the experiment. We think that the equilibration time has little influence on the cells' susceptibility to antibiotics. The reason for that is that the cells loaded into the channels are mid-exponential phase bacteria, and the loading process is short compared to known transition times to stationary phase. During the first 30 minutes, cells already exhibit "robust" growth. It is important to consider that within cell growth is highly dynamic (Fig. 3, SI Fig.S4).

4. The units of the growth-rates are not clear to me, and it is also not very clear how they are calculated, i.e. log-transformed or linear length-data? Usually, specific growth-rate with units 1/h or 1/min is calculated. Here typical values would be ~0.7 or 0.01, respectively, for one (size) doubling per hour. Authors give values around 1.08 for untreated control, which I assume would translate to growth of 8 % length within 4 minutes, which would translate to a specific growth-rate of 1.15 1/h, i.e. 1.67 (size) doublings/hour or 0.08 (size) doublings/4 minutes. I would recommend authors to clarify to facilitate understanding. E.g. in line 205 authors refer to this more generally used terminology.

Response: Thank you for bringing this to our attention. In the new version of the manuscript, we provide more comparable growth rate reporting (e.g. Line 206), ensuring ease of understanding for readers. We still like to highlight that scaling such mean estimates might not be very accurate, as slowly growing cells or shrinking cells do not follow exponential growth patterns. Hence, rates per 4 min, as we measured, provide the most accurate representation.

5. L317: The authors bring up that cells which grow less robust exhibit higher mortality. Have the authors looked at whether variability in growth rate per cell prior to antibiotic exposure would be a good predictor for survival?

Response: This is an insightful suggestion. We explore this possibility in the new version of the article and in the SI (Fig. S4). As you will see growth dynamics are overwhelming any potential predictive signal on survival.

6. Figure 3a (this seems to be confused with figure 2, see comment 11 below): Why are so many cells dying in the absence of antibiotics? It seems like ~90% of the population is lost over less than 24 hours. Is this actual death or is it related to cells escaping the mother machine channels? Authors mention in L198ff that these death rates are expected, quoting their own work. But how do these high death values relate to e.g. the work by Jun group which introduced the mother machine in 2010 (Wang et al, 2010, Curr. Biol.)? They show an average life span of 100 - 150 generations. I would appreciate if this could be clarified.

Response: We also appreciate this question. The observed high death rates in the absence of antibiotics may be related to various factors, including cells escaping from the mother machine, which mainly happens when they filament. Fig. 3 includes such right censored (mother cells that got washed out of their channels, see added information in Fig. 3 legend). Also, exposure to high energy light for fluorescent imaging influences mortality, as does the media (Line 173). The Jun lab uses for instance rich media (LB) while our experiments are done under supplemented M9. Filamentation behaviour under LB is substantially different than under M9. We highlight that our results are

dependent on the condition we use, but refrained from too detailed discussions of all the parameters that can influence overall mortality as our focus was on comparisons among treatment groups and exposure periods.

7. L352-354: *The authors state this as a fact, although this is hypothetical, even though I agree on the potential implication for resistance evolution.*

Response: We have toned down this statement and provide additional references from other studies (Lines 322-340).

8. L354: *It is not clear how the authors arrive at the 4-fold higher potential without referring to the SI.*

Response: This is coming from the number of divisions phenotypic resistant cells have compared to susceptible cells. We elaborate on this and explain how division times should link to mutation probabilities in the revised manuscript (Lines 326-331). We also highlight that other aspects could impact mutation rates (please also see response to the other referee above).

9. *Check use of resistance vs resistant throughout the manuscript, e.g. L25: phenotypic 'resistance cells' to 'resistant cells', also L88*

Response: This has been corrected across the entire manuscript.

10. L103: *arouse is used instead of arose/arise*

Response: Thanks for spotting this, it has been corrected.

11. L181-190: *Figure references are off. It seems like figure 2 and 3 were confused in the submission system.*

Response: We apologize for this confusion.

12. L187-188: *This sentence could require some context, e.g. referring the reader to the discussion section.*

Response: Thanks for the suggestion, we added such context for better guidance of the reader (Lines 158-159).

13. L382: *Authors argue that reduced growth-rate lowers 'general susceptibility' after showing that there is no evidence for this in their data.*

This is the general assumption in the literature. See for example: --

<https://www.pnas.org/doi/full/10.1073/pnas.1509743112>

Response: We elaborate on this point, cite the reference suggested, and emphasize more explicitly on the surprising and contrasting findings we observe compared to the general assumption (for which there is good evidence) in the literature (Lines 356-379).

14. L385: *colanic acid*, not *colonic*

Response: This has been corrected.

15. Figure 2: Units of Age on x-axis (I assume minutes)

Response: We added the label accordingly.

16. Figure 5: Here cells are divided into low, medium and high, whereas in the text they are referred to as growth-arrested, slow-growing and fast-growing (starting from L242).

Response: We adjusted the legend labels.

Re: mSystems00256-24R1 (Phenotypic resistant single-cell characteristics under recurring ampicillin antibiotic exposure in *Escherichia coli*.)

Dear Dr. Ulrich Karl Steiner:

Your manuscript has been accepted, and I am forwarding it to the ASM production staff for publication. Your paper will first be checked to make sure all elements meet the technical requirements. ASM staff will contact you if anything needs to be revised before copyediting and production can begin. Otherwise, you will be notified when your proofs are ready to be viewed.

Sincerely,
Li Cui
Editor
mSystems